# Neofunctionalization of a second insulin receptor gene in the wing-dimorphic planthopper, *Nilaparvata lugens*

Wen-Hua Xue[1], Nan Xu[1], Sun-Jie Chen[1], Xin-Yang Liu[1], Jin-Li Zhang[1], Hai-Jun Xu[1,2,3]*

1 Institute of Insect Sciences, Zhejiang University, Hangzhou, China, 2 State Key laboratory of Rice Biology, Zhejiang University, Hangzhou, China, 3 Ministry of Agriculture Key laboratory of Molecular Biology of Crop Pathogens and Insect Pests, Zhejiang University, Hangzhou, China

* haijunxu@zju.edu.cn

**Data Availability Statement:** The raw data from the RNA-seq on wingbuds and female adults were submitted to GenBank with SRA accession numbers PRJNA675314 and PRJNA724037.

## Abstract

A single insulin receptor (*InR*) gene has been identified and extensively studied in model species ranging from nematodes to mice. However, most insects possess additional copies of *InR*, yet the functional significance, if any, of alternate *InR*s is unknown. Here, we used the wing-dimorphic brown planthopper (BPH) as a model system to query the role of a second *InR* copy in insects. *NlInR2* resembled the BPH *InR* homologue (*NlInR1*) in terms of nymph development and reproduction, but revealed distinct regulatory roles in fuel metabolism, lifespan, and starvation tolerance. Unlike a lethal phenotype derived from *NlInR1* null, homozygous *NlInR2* null mutants were viable and accelerated DNA replication and cell proliferation in wing cells, thus redirecting short-winged–destined BPHs to develop into long-winged morphs. Additionally, the proper expression of *NlInR2* was needed to maintain symmetric vein patterning in wings. Our findings provide the first direct evidence for the regulatory complexity of the two *InR* paralogues in insects, implying the functionally independent evolution of multiple *InR*s in invertebrates.

## Author summary

The highly conserved insulin/insulin-like growth factor signaling pathway plays a pivotal role in growth, development, and various physiological processes across a wide phylogeny of organisms. Unlike a single *InR* in the model species such as the fruit fly *Drosophila melanogaster* and the nematode *Caenorhabditis elegans*, most insect lineages have two or even three *InR* copies. However, the function of the alternative *InR*s remains elusive. Here, we created a homozygous mutation for a second insulin receptor (*InR2*) in the wing-dimorphic brown planthopper (BPH), *Nilaparvata lugens*, using the clustered regularly interspaced palindromic repeats/CRISPR-associated (CRISPR/Cas9) system. Our findings revealed that *InR2* possesses functions distinct from the BPH *InR* homologue (*NlInR1*), indicating that multiple *InR* paralogues may have evolved independently and may have functionally diversified in ways more complex than previously expected in invertebrates.

Sequences of NlInR1 and NlInR2 are available with GenBank accession numbers KF974333 and KF974334.

**Funding:** This work was supported by grants from National Natural Science Foundation of China (Grants 31522047, 31772158, and 31972261 to HJX) and China postdoctoral science foundation (Grant 2020M671739 to JLZ). The funders had no role in study design, data collection and analysis, decision to publish, or preparation of the manuscript.

**Competing interests:** The authors have declared that they have no competing interests exist.

## Introduction

The highly conserved insulin/insulin-like growth factor (IGF) signaling (IIS) pathway is well established as a critical regulator of growth, development, and various physiological processes, including metabolic homeostasis, lifespan, reproduction, and stress responses, across a wide phylogeny of organisms, ranging from nematodes to humans [1–7]. The IIS cascade is activated upon ligand binding, and the actions of insulin and insulin-like growth factors (IGFs) in mice and humans are mediated by insulin receptor (InR) and IGF-1 receptor (IGF-1R), respectively, two distinct transmembrane tyrosine kinases [8,9]. Mice lacking the *InR* gene were born at term with slight growth retardation and died of ketoacidosis soon after birth [10], while mice lacking *Igf-1r* died at birth because of respiratory failure [11,12]. However, a single *InR* but not *Igf-1r* was identified in the fly *Drosophila melanogaster* and in the nematode *Caenorhabditis elegans* [9,13–16]. Genetic evidence derived from targeted *Drosophila* mutants indicated that the allele *Drosophila InR* (*dInR*) mutants were recessive embryonic or early larval lethal, although some heteroallelic complementations of *dInR* alleles were viable and yielded adults with severe developmental delay, reduced body and organ sizes, extended adult longevity, and female sterility [13,17–19]. The effect of *dInR* on the reduction of body and organ size in *Drosophila* was primarily mediated through reduced cell size and number [17,18,20]. In *C. elegans*, *InR* homologue (*daf2*) mutants showed arrested development at the dauer larval stage and increased longevity [15,21]. In addition, InRs have been implicated in the regulation of phenotypic plasticity in some insects. Male rhinoceros beetles (*Trypoxylus dichotomus*) wield a forked horn on their heads with hyper-variability in its size ranging from tiny bumps to exaggerated structures two-thirds the length of a male's body. Knockdown of *T. dichotomus InR* homologue induced a major decrease in the size of the horns [22]. The damp-wood termite (*Hodotermopsis sjostedti*) exhibits various morphological castes associated with the division of labor within a colony. Knockdown of *H. sjostedti InR* homologue in pre-soldier termites disrupted soldier-specific morphogenesis including mandibular elongation [23,24]. These lines of findings provide insights into our understanding of the functional diversity of the conserved *InR* in invertebrates.

In contrast to the single *InR* gene in *Drosophila*, two or even three *InR* copies are conserved in most insect lineages [25–28]. Analysis of the *InR* sequences in 118 insect species from 23 orders indicated that this *InR* multiplicity might result from duplication of the *InR* gene prior to the evolution of flight, followed by multiple secondary losses of one *InR* paralogue in individual lineages, such as in most Diptera [28]. Multiple *InR* paralogues might have distinctly functional implications throughout the life cycle in some insects. The linden bug *Pyrrhocoris apterus* (Hemiptera: Pyrrhocoridae) encoded *InR1a*, *InR1b* and *InR2* paralogues, of which *InR1a* was probably originated through reverse transcription of *InR1b* [28]. Knockdown of *InR1a* or *InR2* in *P. apterus* nymphs led to long-winged (LW) adults; whereas knockdown of *InR1b* led to short-winged (SW) adults [28]. In addition, accumulated evidences indicated that multiple *InRs* were caste-specifically expressed in the honey bee *Apis mellifera* [29], bumblebee *Bombus terrestris* [30], and fire ant *Solenopsis invicta* [31], suggesting that InRs might be involved into caste polyphenism of social insects. Furthermore, in addition to functional discrepancy, multiple *InR* paralogues may have overlapping functions on some aspects of life-history traits in some insects. Knockdown of either two *InR* paralogues impaired fecundity in the green lacewing *Chrysopa pallens* [32] and red flour beetle *Tribolium castaneum* [33], and disrupted nymph-adult transition in the brown citrus aphid *Aphis* (*Toxoptera*) *citricidus* [34]. These evidences raise an intriguing question regarding to the extent of functional conservation between multiple *InR* paralogues in insects.

The wing-dimorphic brown planthopper (BPH), *Nilaparvata lugens* (Hemiptera: Delphacidae), is a classic and representative example of wing polyphenism in insects [35]. BPH nymphs

can develop into SW or LW adults in response to environmental cues, the former have fully developed wings and functional indirect flight muscles (IFM), while the latter exhibit reduced wings and underdeveloped IFM. Although persuasive direct evidence is lacking, juvenile hormone (JH) has long been considered to be the main subject of endocrine regulation of wing polyphenism in BPH as well as in several other wing-polyphenic insects [35–37]. Topical application of JH and its agonists at certain juvenile stages could significantly decrease the percentage of LW morphs in BPH [38–40], the aphid *Aphis fabae* [41], and the cricket *Gryllus rubens* [42,43]. In contrast, treatment with a JH antagonist (precocene II) induced LW morphs in SW BPH population [44,45].

Recently, genetic analysis showed that the activity of IIS cascade determines alternative wing morphs in BPH. BPH has four insulin-like peptides and two *InR* paralogues, *NlInR1* and *NlInR2*, and they share a high sequence similarity and resemble domain structures. However, only *NlInR1* was functionally analogous to *dInR* with respect to development, metabolic homeostasis, stress responses, lifespan, reproduction, and starvation tolerance [25,46,47]. Activation of *Nl*InR1 activates of the phosphatidylinositol-3-OH kinase [*Nl*PI(3)K]-protein kinase B (*Nl*Akt) signaling cascade, which in turn inactivates the forkhead transcription factor subgroup O (*Nl*FoxO), thus leading to the LW morph. However, *Nl*InR2 can antagonize the *Nl*InR1 activity, and as a result activates the *Nl*FoxO activity, leading to the SW morph [46]. Hence, RNA interference (RNAi)-mediated silencing of *NlInR1* (*NlInR1*[RNAi]) and *NlInR2* (*NlInR2*[RNAi]) led to SW and LW, respectively, through common signaling elements of the *Nl*PI(3)K-*Nl*Akt-*Nl*FoxO cascade [46]. This finding provides an additional layer of regulatory mechanism underlying wing dimorphism in BPHs.

Here, we aimed to elucidate the function of a second *InR* copy in insects by creating loss-of-function *NlInR2* mutations in *N. lugens* using the clustered regularly interspaced palindromic repeats/CRISPR-associated (CRISPR/Cas9) system. We demonstrated that *NlInR2* shared analogous functions with *NlInR1* in terms of organism development and fertility, but differed in the effects on fuel metabolism, adult lifespan, starvation tolerance, and tissue growth. Furthermore, *NlInR2* might play an important role in symmetrical patterning of wing veins. These findings provide the first direct evidence of distinct functions for the two *InR* paralogues in insects, and thus further our understanding of the evolution of InRs in invertebrates.

## Results

### *NlInR2*-null mutants develop into viable long-winged morphs

*Nl*InR2 and *Nl*InR1 closely resemble each other as well as their *P. apterus* counterparts with respect to domain architecture and amino acid similarity (S1 Fig). The intron-exon structure of the *NlInR2* gene was deduced by comparison of cDNA (GenBank accession number: KF974334) and genomic sequences. The coding sequence of *NlInR2* was contained within eight exons spanning approximately 940 kilobase pairs of genomic DNA. We designed a single guide RNA (sgRNA) for CRISPR/Cas9-mediated mutagenesis of *NlInR2*, which was located at 36 to 54 nucleotide (nt) downstream of the *NlInR2* start coden (ATG) in exon 4 (Figs 1A and S1). Pre-blastoderm eggs of wild-type (*Wt*) SW BPHs (*Wt*[SW]) were collected for microinjection with a mix of sgRNA and Cas9 mRNA, and these eggs were then reared to adults (G_0). The genotypes of these G_0 BPHs were subsequently determined by Sanger sequencing, and a heterozygous G_0 female with an 11-nt deletion in the vicinity of the Cas9 cleavage site was picked for creation of homozygous *NlInR2*-null mutants (*NlInR2*[E4], Fig 1B). Sanger sequencing indicated that *NlInR2*[E4] BPHs had an 11-nt deletion in exon 4 (Fig 1C), presumably resulting in a frame shift of the coding region of *NlInR2* and a complete dysfunction of *Nl*InR2 protein (S1 Fig). *NlInR2*[E4] BPHs were viable, and had hind tibia length (Fig 1D), head size

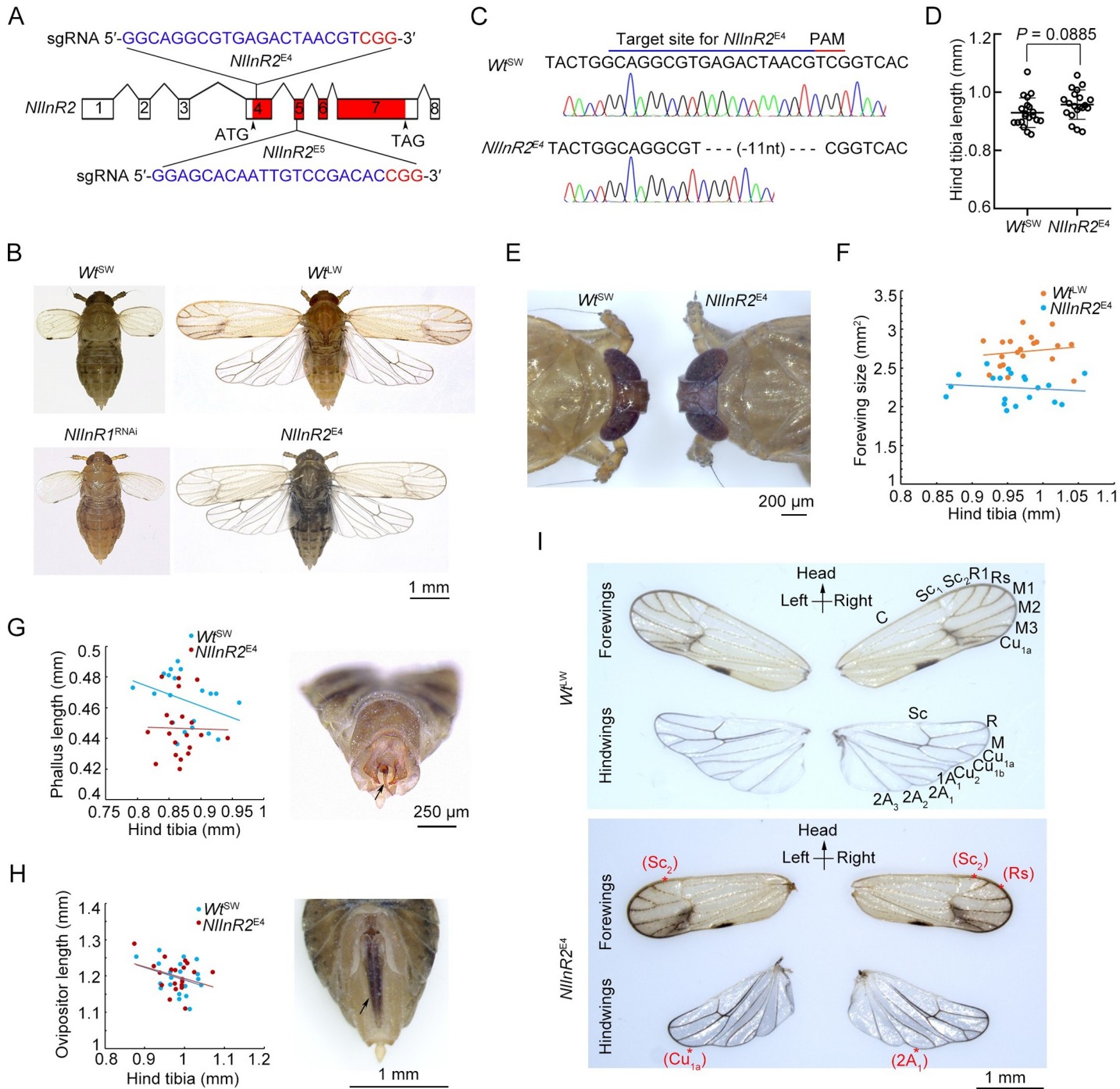

**Fig 1. CRISPR/Cas9-mediated mutations at the *NlInR2* locus.** **(A)** Schematic diagram of sgRNA-targeted sites in exons 4 and 5 of *NlInR2*. Exons composing the *Nl*InR2 cDNA were indicated by numbers, and the encoding region of *NlInR2* was indicted by exons in red. Target sequences for generating *NlInR2* mutations indicated in blue and the PAM indicated in red, which were located at 36 to 54 nucleotide (nt) downstream of the *NlInR2* start coden (ATG) in exon 4. **(B)** Morphologies of BPHs. *Wt*$^{SW}$, wild-type short-winged BPHs; *Wt*$^{LW}$, wild-type long-winged BPHs; *NlInR1*$^{RNAi}$, RNAi-mediated knockdown of *NlInR1* in *Wt*$^{SW}$ BPHs; *NlInR2*$^{E4}$, homozygous *NlInR2*-null mutants derived from *Wt*$^{SW}$ BPHs. **(C)** Sanger sequencing of region flanking target sites in *NlInR2*$^{E4}$ BPHs. Exon 4 of *NlInR2*$^{E4}$ locus had an 11-nt deletion. **(D)** Hind tibia length of *Wt*$^{SW}$ ($n = 20$) and *NlInR2*$^{E4}$ ($n = 20$) females. Statistical comparisons was performed using a two-tailed Student's *t*-test (**, $P < 0.01$ and ****, $P < 0.0001$), and bars represent mean ± s.e.m. **(E)** Head sizes of *Wt*$^{SW}$ and *NlInR2*$^{E4}$ females. **(F)** Relative forewing size and hind tibia length in *Wt*$^{SW}$ ($n = 20$) and *NlInR2*$^{E4}$ ($n = 20$) females. **(G)** Relative phallus and hind tibia length in *Wt*$^{SW}$ ($n = 20$) and *NlInR2*$^{E4}$ ($n = 20$) males. The morphology of male external genitalia was shown on the right and the phallus was indicated by an arrow. **(H)** Relative ovipositor and hind tibia length in *Wt*$^{SW}$ ($n = 20$) and *NlInR2*$^{E4}$ ($n = 20$) females. The morphology of female external genitalia was shown on the right and the ovipositor was indicated by an arrow. Each dot in **(D)**, **(F)**, **(G)**, and **(H)** represents the forewing size derived from an individual female. **(I)** Vein patterning in forewings and hindwings of *NlInR2*$^{E4}$ and *Wt*$^{LW}$ BPHs. Normal veins were labeled in *Wt*$^{LW}$ forewings and hindwings. Missing veins in forewings (Sc$_2$ or Rs) and hindwings (Cu$_{1a}$ or 2A$_1$) of *NlInR2*$^{E4}$ indicated by stars.

(Fig 1E) and compound eyes (S2 Fig) comparable to $Wt^{SW}$, indicating that $NlInR2^{E4}$ had a similar body size to $Wt^{SW}$ controls. Notably, all $NlInR2^{E4}$ adults were LW morphs and were thus morphologically distinct from $Wt^{SW}$ controls and $NlInR1^{RNAi}$ BPHs in terms of wing size (Fig 1B), but similar to $Wt$ LW BPHs ($Wt^{LW}$, Fig 1B). The $NlInR1^{RNAi}$ adults were derived from ds$NlInR1$-treated 4th-instar nymphs and had ~67% decreased expression of $NlInR1$ relative to $Wt^{SW}$ (S3 Fig). Morphometric measurements on forewings of $NlInR2^{E4}$ showed that the size 17% smaller than that of $Wt^{LW}$ (Fig 1B and 1F). In addition, $NlInR2^{E4}$ slightly and significantly reduced phallus length (Fig 1G), but had ovipositors with the size comparable to $Wt^{SW}$ (Fig 1H). Notably, we noticed that the correlation coefficients ($R^2$) of wing size (Fig 1F), phallus length (Fig 1G), and ovipositor length (Fig 1H) relative to its hind tibia length in $Wt^{LW}$ and $NlInR2^{E4}$ BPHs were no more than 0.13 each, indicating that the sizes in these tissues were not correlated with the body size. These observations indicate that both wings and male genitalia might be more sensitive to $Nl$InR2 activity than eyes and legs in BPH.

It bears emphasizing that majority of $NlInR2^{E4}$ BPHs (70%, $n = 40$) lost one or two wing veins in forewings and hindwings, leading to a single individual with different wing patters at both sides of its body. One forewing of a $NlInR2^{E4}$ BPH lacked the $Sc_2$ vein, but the other lacked both $Sc_2$ and Rs veins (Fig 1I), or the Rs vein was incorrectly positioned. For hindwings of a $NlInR2^{E4}$ BPH, one lost the $Cu_{1a}$ vein, whereas the other was deficient in the $2A_1$ vein (Fig 1I). In contrast, only a few $Wt^{LW}$ BPHs (5%, $n = 20$) had wings with asymmetric vein patterns, and most (95%, $n = 20$) had normally patterned and symmetric forewings and hindwings (Fig 1I). To rule out the possibility of off-target mutagenesis, we generated a second homozygous $NlInR2$ mutant ($NlInR2^{E5}$) by targeting mutagenesis in exon 5 using CRISPR/Cas9 (Figs 1A and S1). $NlInR2^{E5}$ and $NlInR2^{E4}$ BPHs were morphologically identical, implying that the LW morph and asymmetric wing pattern were authentically caused by the loss-of-function $NlInR2$ mutation.

## $NlInR2$ differs from $NlInR1$ in life-history traits

Except for 4th-instar stage, $NlInR2^{E4}$ mutants significantly prolonged each developmental stages compared with $Wt^{SW}$ controls (Fig 2A), thus extending the total nymphal duration from 18 to 21 days. Newly emerged $NlInR2^{E4}$ adults showed slightly but significantly reduced body weights compared to $Wt^{SW}$ controls as well as $Wt^{LW}$ controls since $Wt^{LW}$ controls have a greater body weight than $Wt^{SW}$ controls (Fig 2B). However, $NlInR2^{E4}$ adults had comparable glucose (Fig 2C) and triglyceride (Fig 2D) contents compared to $Wt^{SW}$ controls. In addition, analogous to $NlInR1^{RNAi}$ [47], $NlInR2^{E4}$ females showed ~27% reduction in fecundity compared to $Wt^{SW}$ controls (Fig 2E) although $NlInR2^{E4}$ females had well-developed ovaries morphologically similar to those in $Wt^{SW}$ (S4 Fig), which were in stark contrast to immature ovaries in $NlInR1^{RNAi}$-treated females. In addition, the correlation coefficients between egg numbers and hind tibia length in $Wt^{LW}$, $Wt^{SW}$, and $NlInR2^{E4}$ were significantly low ($< 0.009$, Fig 2E), indicating that fecundity may be not correlated with its body size in BPH. Moreover, western blot analysis showed that $NlInR2^{E4}$ had a comparable level of vitellogenin (Vg) in ovaries relative to $Wt^{SW}$ (S4 Fig). These observations indicate that the fecundity defect in $NlInR2^{E4}$ was not likely due to Vg expression. Notably, $Wt^{SW}$ and $Wt^{LW}$ controls laid comparable numbers of eggs although they are morphologically different in wing size (Fig 2E).

In contrary to extended adult lifespan and increased starvation tolerance derived from $NlInR1^{RNAi}$ [47], depletion of $NlInR2$ had a marginal effect on adult lifespan (Fig 2F) and significantly reduced starvation tolerance (Fig 2G). Taken together, our findings indicate that $Nl$InR2 resembles $Nl$InR1 on nymphal development and fecundity, but differs from $Nl$InR1 on fuel metabolism, lifespan, and starvation tolerance.

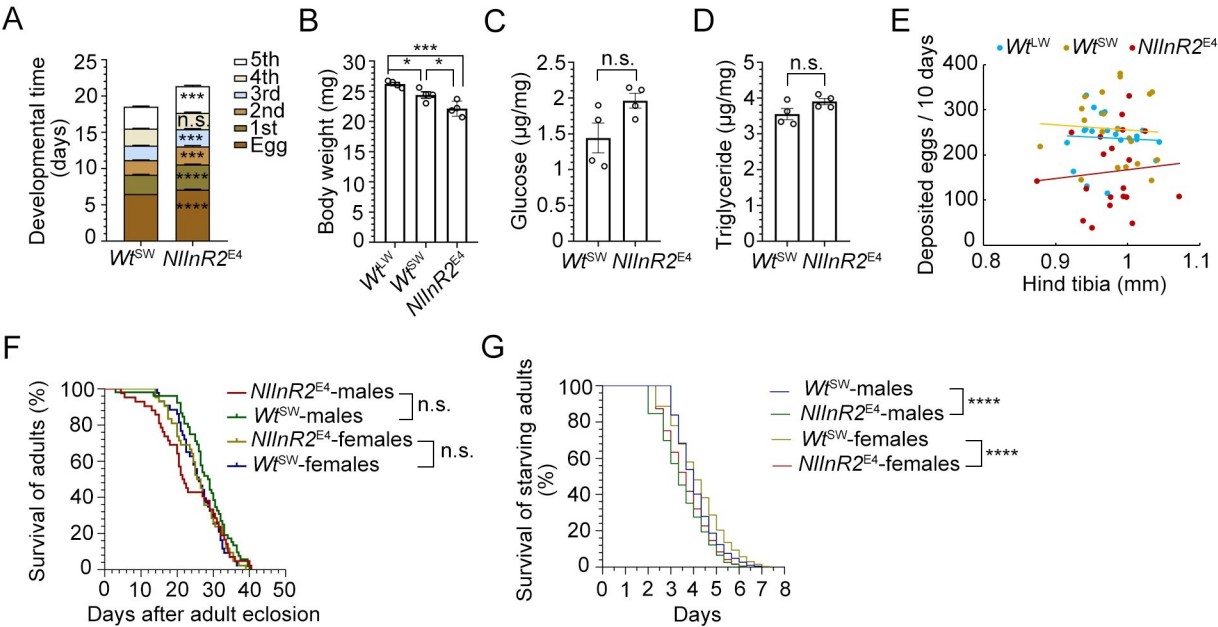

**Fig 2. Life-history traits of *NlInR2*-null mutants. (A)** Duration of developmental stages of *NlInR2*[E4] and *Wt*[SW]. Data are presented as mean ± s.e.m for 20 independent biological replicates (*n* = 20). Two corresponding columns were compared using two-tailed Student's *t*-test (***, *P* < 0.001; ****, *P* < 0.0001). **(B, C,** and **D)** *NlInR2*[E4] and *Wt*[SW] females at 12 hAE were pooled to measure boy mass **(B)**, glucose content **(C)**, and triglyceride content **(D)**. Each circle represents each sample pooled from 15 females. Bars represent mean ± s.e.m. derived from four independent biological replicates. Two groups were compared using two-tailed Student's *t*-test (*, *P* < 0.05; ***, *P* < 0.001; n.s., no significance). **(E)** Relative fecundity and hind tibia length in*NlInR2*[E4] and *Wt*[SW] BPHs. One female was paired with two males, and then allowed to lay eggs for 10 days. Each circle represents eggs produced by an individual female (*n* = 20). Hind tibia length was used to represent body size. **(F)** Survival rates of *NlInR2*[E4] and *Wt*[SW] adults. Newly emerged *NlInR2*[E4] BPH (*n* = 42 females, and *n* = 42 males) and *Wt*[SW] BPHs (*n* = 43 females and *n* = 52 males) were collected for survival assay. No significant (n.s.) difference was found between *NlInR2*[E4] and *Wt*[SW] BPHs (log-rank Mantel-Cox test). **(G)** Starvation tolerance assays of *NlInR2*[E4] and *Wt*[SW] adults. Females (*n* = 30) and males (*n* = 30) at 24 h after eclosion were fed with water only for starvation assays. Statistical analysis was performed by log-rank Mantel-Cox test (****, *P* < 0.0001).

### *NlInR2*-null mutants increase wing cell number

To look into further how the LW phenotype developed in *NlInR2*[E4] mutants, we investigated wing-cell numbers in 5th-instar nymphs by quantifying genomic DNA copy number using quantitative real-time PCR (qRT-PCR). Because wing buds grow explosively during an approximately 48 h time window at the beginning of the 5th-instar nymph stage [48], fifth-instar nymphs were collected at 24 h intervals after ecdysis, and the wing buds were dissected from the second thoracic segment (T2W) and third thoracic segment (T3W) (Fig 3A). The number of cells in T2W from 5th-intar *NlInR2*[E4] nymphs at 3 h after ecdysis (hAE) were comparable to that from *Wt*[SW] nymphs (Fig 3B). During 24–48 hAE, *NlInR2*[E4] T2W proliferated at a higher rate than *Wt*[SW] T2W and reached to a maximum at 48 hAE, although both *NlInR2*[E4] and *Wt*[SW] T2Ws began to accumulate cells at same time (24 hAE) (Fig 3B). A similar phenotype was observed in T3W, except that *NlInR2*[E4] T3W began to proliferate before 24 hAE, and the cell number in *Wt*[SW] T3W remained unchanged for the first 48 h, and then decreased at 72 hAE (Fig 3C).

To further confirm that the *NlInR2*-null mutation increased cell proliferation in the wings, we microinjected 24h-5th-instar *NlInR2*[E4] and *Wt*[SW] nymphs with 5-ethynyl-2′-deoxyuridine (EdU). EdU is a thymidine analogue in which a terminal alkyne group replaces the methyl group in the 5 position, and is readily incorporated into cellular DNA during DNA replication [49]. T2W and T3W were dissected from nymphs at 24 h after microinjection for EdU

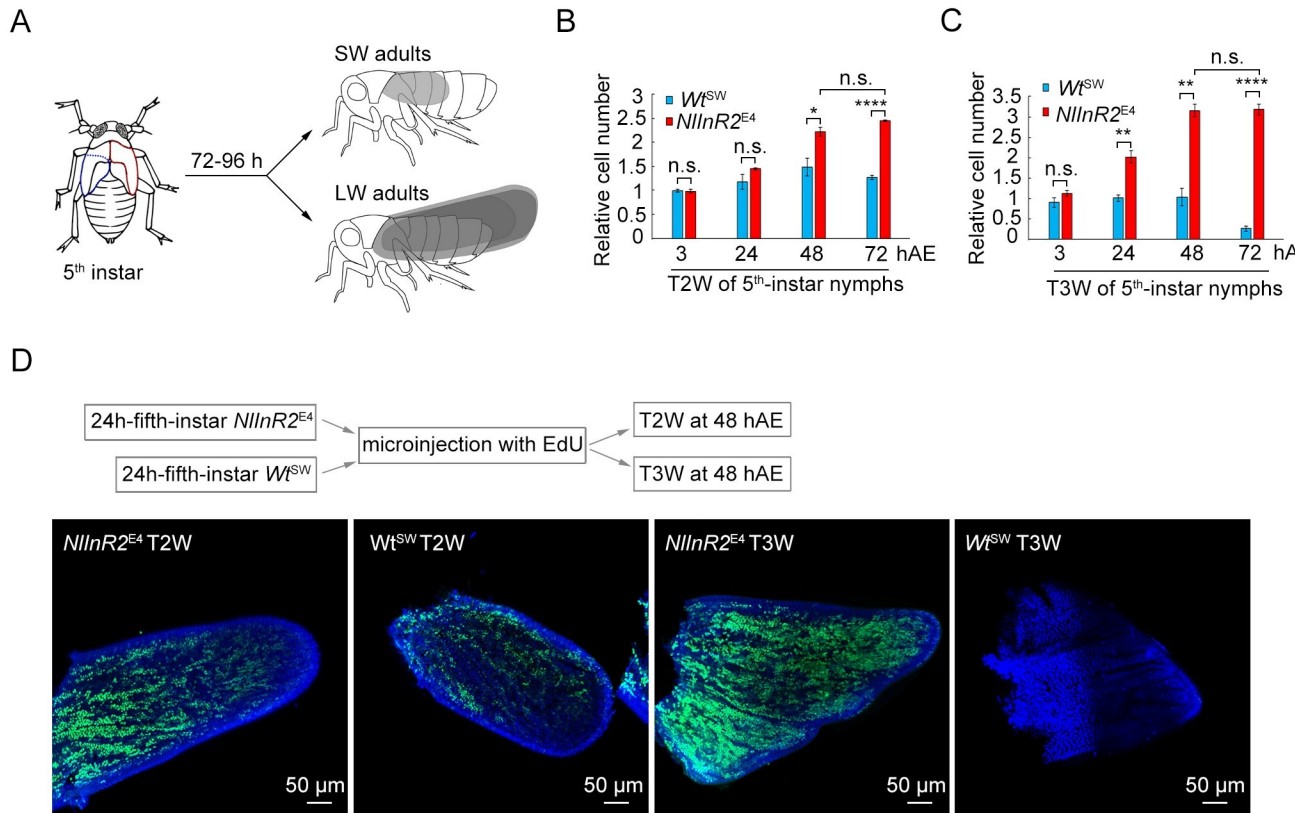

**Fig 3. *NlInR2*-null mutants increase wing cell number. (A)** Schematic diagram of development of alternative wing morphs in BPHs. Fifth-instar nymphs can molt into short-winged (SW) or long-winged (LW) adults. Wing buds in the second thoracic segment (T2W) and third thoracic segment (T3W) indicated by red and blue dots, respectively. **(B** and **C)** Relative wing cell numbers in 5th-instar *NlInR2*E4 and *Wt*SW nymphs. T2W **(B)** and T3W **(C)** were dissected from 150 nymphs at 24-, 48, or 72 h after ecdysis (hAE), and genomic DNA was isolated for qRT-PCR analysis. Bars represent mean ± s.e.m. derived from three independent biological replicates. Statistical comparisons between two columns were performed using two-tailed Student's *t*-test (*, *P* < 0.05; **, *P* < 0.01; ****, *P* < 0.0001; n.s., no significance). **(D)** EdU staining of newly proliferated cells in wing buds during 24–48 hAE. Fifth-instar *NlInR2*E4 and *Wt*SW nymphs at 24 hAE were microinjected with EdU, and T2W and T3W were dissected for Hoechst33342 (in blue) and EdU (in red) staining at 48 hAE.

staining. *NlInR2*E4 T2W had more EdU-stained cells than *Wt*SW T2W (Fig 3D), indicating that genomic DNA replicated more frequently in *NlInR2*E4 T2W. However, EdU was barely detected in *Wt*SW T3W, in contrast to the strong signals in *NlInR2*E4 T3W (Fig 3D). These observations are in accord with the increased cell numbers in *NlInR2*E4 wings detected by qRT-PCR, suggesting that LW development in *NlInR2E4* BPHs is likely caused by an accelerated cell proliferation rate.

## Knockdown of *NlInR1* compromises effects of *NlInR2*E4 on long wing and indirect flight muscles development

Given that *NlInR1* and *NlInR2* demonstrated opposite roles in the formation of alternative wing morphs [46], we determined if dysfunction of *NlInR1* could abolish long wing development in the context of *NlInR2*–null mutation. For this purpose, we microinjected 12h-5th-instar *NlInR2*E4 nymphs with ds*NlInR1* (*NlInR2*E4;ds*NlInR1*) or ds*Gfp* (*NlInR2*E4;ds*Gfp*). The *NlInR1* expression decreased to ~38% in *NlInR2*E4;ds*NlInR1* BPHs relative to *NlInR2*E4;ds*Gfp* BPHs (S3 Fig). In contrast to 100% LW adults derived from *NlInR2*E4;ds*Gfp*, knockdown of *NlInR1* in *NlInR2*E4 caused most female (Pearson's $\chi^2$ test: $\chi^2$ = 110, *df* = 1, *P* < 0.0001) and male (Pearson's $\chi^2$ test: $\chi^2$ = 99.733, *df* = 1, *P* < 0.0001) nymphs to develop into SW morphs

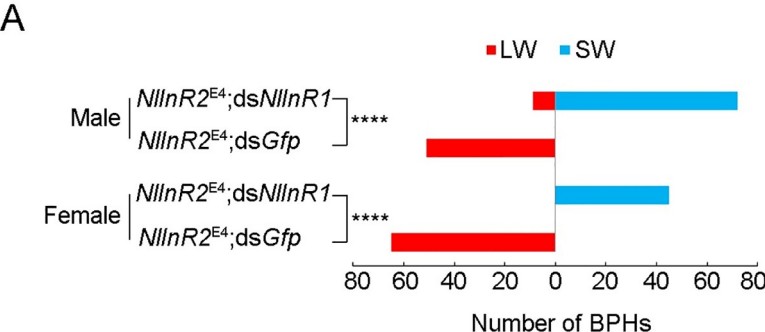

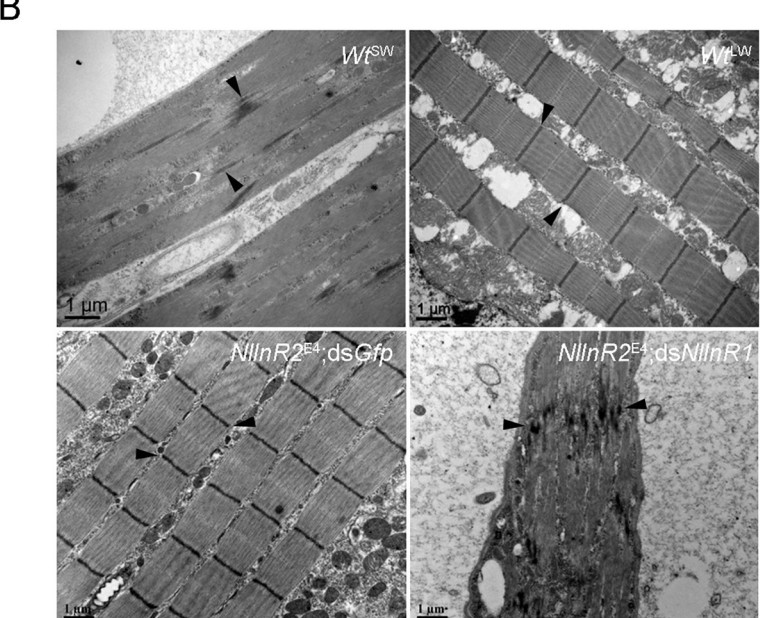

**Fig 4. Knockdown of *NlInR1* compromises *NlInR2*-null mutants in terms of LW and indirect flight muscles development.** Fifth-instar *NlInR2*[E4] nymphs at 12 hAE were microinjected with dsRNAs targeting *NlInR1* (*NlInR2*[E4]; ds*NlInR1*) or *Gfp* (*NlInR2*[E4];ds*Gfp*). Long-winged (LW) and short-winged (SW) adults were counted after adult eclosion (**A**). Groups were compared Pearson's $\chi^2$ test (****, $P < 0.0001$). Thoraxes were dissected from 24h-female adults for indirect flight muslces examination by transmission electron microscopy (**B**). *Wt*[SW], wild-type short-winged BPHs. *Wt*[LW], wild-type long-winged BPHs. The Z discs of indirect flight muscles indicated by arrowheads.

([Fig 4A]). Furthermore, transmission electron microscopy (TEM) showed that *NlInR2*[E4]; ds*NlInR1*-treated BPHs had degenerated indirect flight muscles and distorted Z discs in the sarcomere, thus resembling *Wt*[SW] BPHs ([Fig 4B]). In contrast, indirect flight muscles in *NlInR2*[E4];ds*Gfp*-treated BPHs was well-organized, as for *Wt*[LW] BPHs ([Fig 4B]). These results indicate that the early 5th-instar stage is a critical window for wing-morph development, during which stage *NlInR1* knockdown could antagonize the effects of *NlInR2* knockout on wing and indirect flight muscles development.

## *NlInR2*-null mutants temporally up-regulate genes associated with DNA replication in wings of 5th-instar nymphs

To clarify the molecular basis underlying LW development in *NlInR2*[E4] BPHs, we carried out comparative transcriptomic analysis of T2W and T3W of 5th-instar *NlInR2*[E4] and *Wt*[SW]

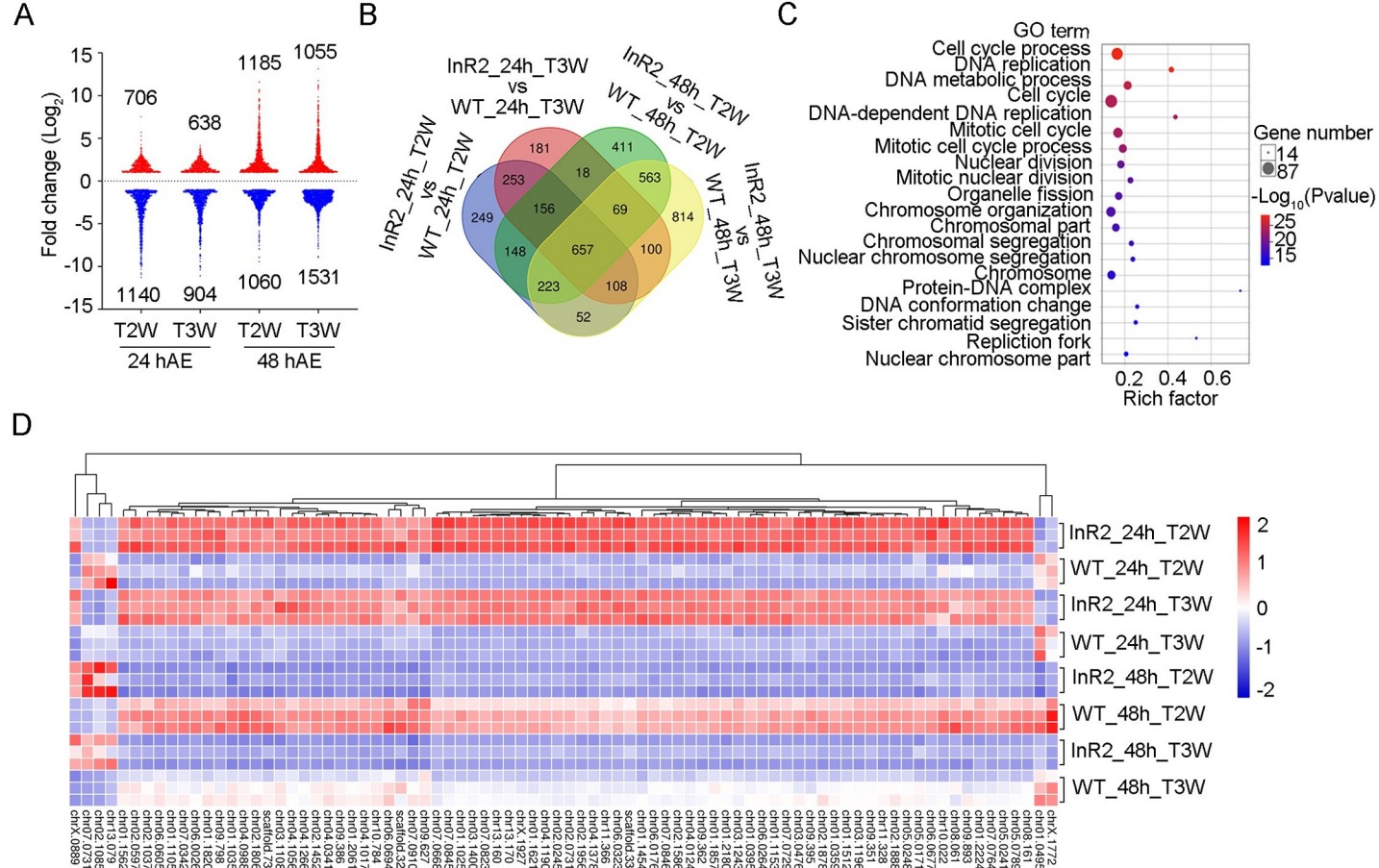

**Fig 5. Comparative transcriptomic analysis of wing buds from 5th-intar *NlInR2*^E4 and *Wt*^SW nymphs. (A)** Differentially expressed genes (DEGs) in wing buds from 5th-instar *NlInR2*^E4 and *Wt*^SW nymphs. Numbers of up-regulated and down-regulated genes are indicated. T2W and T3W represent wing buds on the second and third thoracic segment, respectively. DEGs were screened based on the criteria of fold-change ≥ 2 and adjusted *P* (padj) < 0.01. 24 and 48 hAE indicate 5th-instar nymphs at 24 and 48 h after ecdysis, respectively. **(B)** Venn diagram of common/specific DEGs in wing buds of 5th-instar *NlInR2*^E4 and *Wt*^SW nymphs. A total of 657 DEGs were commonly regulated by T2W and T3W of 5th-instar *NlInR2*^E4 and *Wt*^SW nymphs at 24 and 48 hAE. **(C)** The top 20 enriched Gene Ontology (GO) terms of commonly regulated DEGs (657) in wing buds of *NlInR2*^E4 and *Wt*^SW nymphs. The GO term of 'cell cycle process' (GO: 0022402) was the most significantly enriched. **(D)** Heatmap of DEGs in the GO term of 'cell cycle process' (GO: 0022402). Up-regulated and down-regulated genes are indicated in red and blue, respectively. Expression level indicated by log2FoldChang. InR2_24h_T2W and WT_24h_T2W: T2W from 24h-5th-instar *NlInR2*^E4 and *Wt*^SW nymphs, respectively. InR2_24h_T3W and WT_24h_T3W: T3W from 24h-5th-instar *NlInR2*^E4 and *Wt*^SW nymphs, respectively. InR2_48h_T2W and WT_48h_T2W: T2W from 48h-5th-instar *NlInR2*^E4 and *Wt*^SW nymphs, respectively. InR2_48h_T3W and WT_48h_T3W: T3W from 48h-5th-instar *NlInR2*^E4 and *Wt*^SW nymphs, respectively.

nymphs at 24- and 48-hAE (S1 Data and S1 Table). *NlInR2*-null mutants had differentially expressed genes (DEGs) accounting for 8.3%–14% of BPH encoding genes (18,534 genes, Fig 5A and S1 Data). In 5th-instar nymphs at 24- or 48-hAE, most of the up-regulated genes in T2W of *NlInR2*^E4 were significantly enriched in Gene Ontology (GO) terms associated with DNA replication and cell proliferation at 24 hAE, but these were down-regulated at 48 hAE (S1 Data and S2–S4 and S8–10 Tables). Intriguingly, the same expression pattern was observed in *NlInR2*^E4 T3W (S1 Data and S5–S7 and S11–S13 Tables).

Additionally, 657 DEGs were commonly regulated by T2W and T3W in *NlInR2*^E4 and *Wt*^SW nymphs at 24- and 48-hAE (Fig 5B). The most significantly enriched GO term (GO: 0022402) was assigned to the terms such as cell cycle process (82 genes) and DNA replication (38 genes, Fig 5C and S14 Table). Most of the genes associated with cell cycle process were

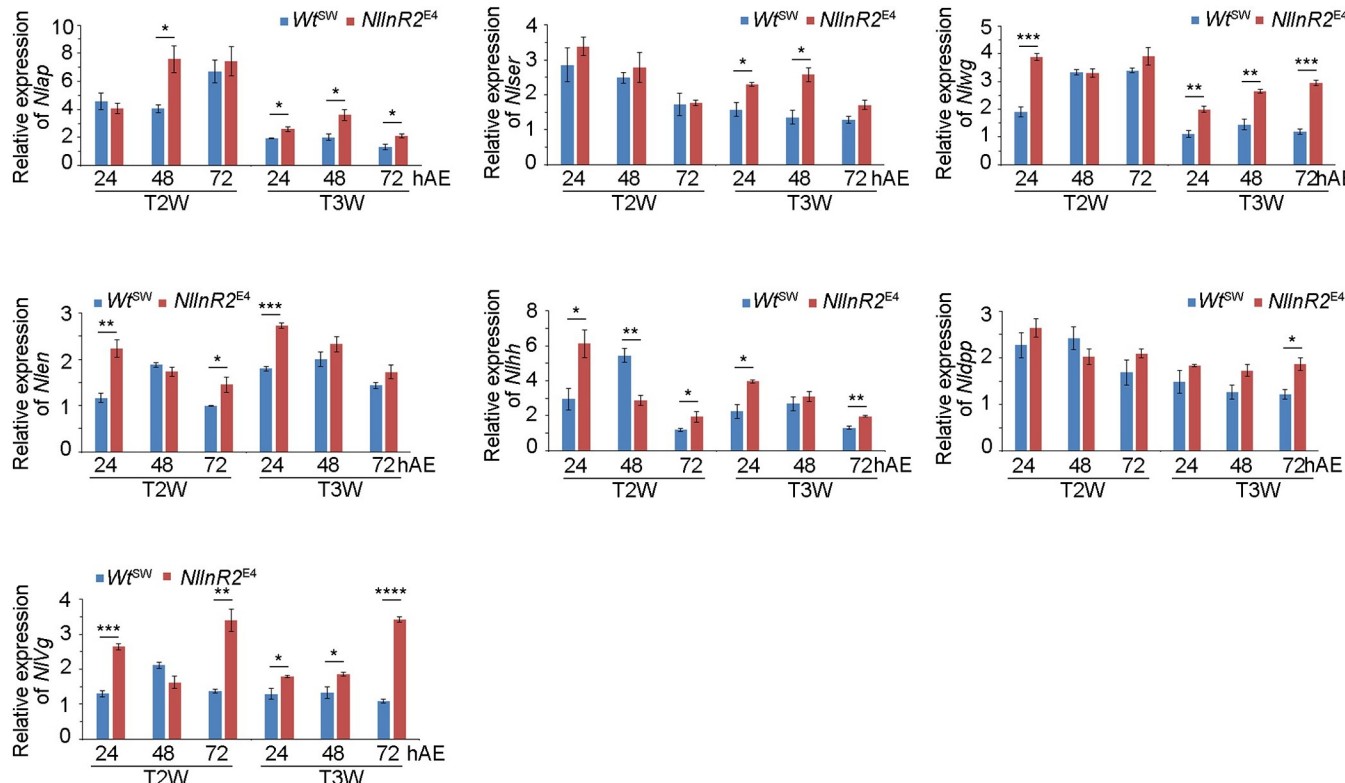

**Fig 6. Expression of wing-patterning genes in NlInR2-null mutants during the 5th-instar stage.** T2W and T3W were dissected from 24, 48, and 72 hAE 5th-instar NlInR2E4 and WtSW nymphs (n = 150), and total RNA was isolated for qRT-PCR. Relative expression of each gene was normalized by rps15. Bars represent mean ± s.e.m. derived from three independent biological replicates. Statistical comparisons between two groups were performed using two-tailed Student's t-test (*, P < 0.05, **, P < 0.01, and ***, P < 0.001). Nlap, N. lugens apterous homologue; Nlser, N. lugens serrate homologue; Nlwg, N. lugens wingless homologue; Nlen, N. lugens engrailed homologue; Nlhh, N. lugens hedgehog homologue; Nldpp, N. lugens decapentapleigic homologue; Nlvg, N. lugens vestigial homologue.

up-regulated in NlInR2E4 compared with WTSW at 24 hAE, but down-regulated at 48 hAE (Fig 5D and S15 Table). Overall, these events indicate that NlInR2E4 might temporally activate LW development by comprehensively regulating a battery of genes involved in DNA replication and the cell cycle.

### NlInR2-null mutants tempo-spatially elevate the expression levels of wing-patterning genes

Wing formation in *Drosophila* depends on signals from both the anterior-posterior (AP) and dorsal-ventral (DV) axes, which are defined by the actions of the engrailed (en) and apterous (ap) selector proteins, respectively. Wing growth and patterning are organized by the morphogens hedgehog (hh), decapentapleigic (dpp), and (wingless) wg secreted from the AP and DV compartment boundaries, respectively, together with numerous downstream wing-patterning genes organize wing growth and patterning [50–54].

To gain insights into how NlInR2 null mutation affected wing formation, we examined the expression level of seven key wing-patterning genes including en, ap, hh, dpp, wg, serrate (ser), and vestigial (vg), in 5th-instar NlInR2E4 and WtSW BPHs at 24-, 48-, and 72-hAE. NlInR2-null mutation slightly but significantly elevated expression levels of all seven genes in a tempo-spatially dependent manner (Fig 6).

## Discussion

Owing to the extensive molecular genetic toolbox, studies of *Drosophila dInR* mutants have contributed greatly to our understanding of the complex regulation of *InR* in the life cycle of a wide variety of insects, and its functional conservation across the animal kingdom. However, in contrast to the single *InR* gene in *Drosophila*, most insect lineages have two or even three *InR* copies, although our understanding of their functions remains limited. Here, we used BPH as a model system to query the role of a second *InR* copy in the life-history traits of insects. Our findings revealed distinct and overlapping functions of *NlInR1* and *NlInR2* in BPH, indicating that multiple *InR* paralogues may have evolved independently and may have biological functions more complex than previously expected.

Although *NlInR1* and *NlInR2* share a high sequence identity [46]; (S1 Fig), only *NlInR1* resembles *Drosophila dInR* in terms of the regulation of growth, development, and a wide spectrum of physiological process. RNAi-mediated silencing of *NlInR1*, but not *NlInR2*, led to dwarf SW BPHs, which exhibited growth retardation, reduced body weight, an extended adult lifespan, an enhanced starvation tolerance, a decreased fertility, and impaired carbohydrate and lipid metabolism [46,47]. In addition, like *Drosophila dInR*, homozygous *NlInR1* mutants were early embryonic lethal, whereas heterozygous mutants resembled *NlInR1*[RNAi] [55]. In the present study, we created viable homozygous *NlInR2*-null mutants by disrupting exons 4 and 5 of *NlInR2* using CRISPR/Cas9. Like *NlInR1*[RNAi], *NlInR2*-null mutants resulted in developmental delay and decreased fertility. However, unlike *NlInR1*[RNAi], *NlInR2*-null mutants had marginal effects on fuel metabolism, lifespan, and decreased starvation tolerance. This observation stands in sharp contrast to the extended longevity of *InR* mutants in major model organisms, including worms [15], flies [19], and mice [56]. Although the exact mechanisms are unknown, this phenotypic feature in *NlInR2* mutants may challenge the evolutionarily conserved roles of *InR* in fuel metabolism and longevity.

One important and seemingly paradoxical difference between *NlInR1* and *NlInR2* was that knockdown of *NlInR1* led to SW BPHs, while *NlInR2*-null mutation induced wing development, leading to LW BPHs. Previous studies in *Drosophila* showed that *dInR*-deficient flies had smaller bodies and organs because of a reduction in both cell size and cell number [17,18,20]. This mechanism may also explain the small body and wing sizes in *NlInR1*[RNAi] BPHs [46]. However, *NlInR2*-null mutation likely increased wing-cell number by accelerating cell proliferation during the first 48 h after 5[th]-instar eclosion. The effect of *NlInR2* on cell proliferation was further evidenced by RNA-seq analysis on wing buds of 5[th]-instar *NlInR2*[E4] nymphs at 24 hAE, which showed that the most up-regulated genes were associated with DNA replication and cell cycle process. One possible explanation for the opposite effects of *NlInR1* and *NlInR2* is that *NlInR2* may serve as a negative regulator of *NlInR1*, as proposed previously [46], and depletion of *NlInR2* thus activates the canonical IIS pathway to stimulate cell proliferation. Intriguingly, RNA-seq on female adults showed that only 884 and 417 genes were differentially expressed by *NlInR1*[RNAi] and *NlInR2*[E4] (S2 Data and S16–S19 Tables), which accounted for 4.8% and 2.2% of BPH encoding genes (18,534), respectively. Moreover, only 101 genes were commonly regulated by *NlInR1*[RNAi] and *NlInR2*[E4]. Thus, this observation is in contrast to the notion that *NlInR2* is a negative regulator of *NlInR1*. Previous studies indicated that *Nl*FoxO is a main IIS downstream effector relaying the *Nl*InR1 and *Nl*InR2 activities. Lin *et al*. recently found that wing cells in SW-destined BPHs were largely in the G2/M phase of the cell cycle, whereas those in LW individuals (*NlFoxO* RNAi) were largely in G1 [57]. This is consistent with the higher cell proliferation rate of *NlInR2*[E4] wing buds in the current study. However, how the single transcription factor *Nl*FoxO exerts different functions in respond to *Nl*InR1 and *Nl*InR2 activities is still an open question.

Previous studies indicated that *NlInR2* was mainly expressed in wing buds [46], which may explain why *NlInR2* null had a marginal effect on body size. Intriguingly, depletion of *Nl*InR2 decreased the phallus length, indicating that male genitalia could respond to the *Nl*InR2 activity. A similar phenomenon was observed in the dung beetle *Onthophagus taurus* when the activity of *O. taurus InR1* homologue was compromised by RNAi knockdown [58]. In addition, *NlInR2*-null mutation tempo-spatially elevated expression levels of wing-patterning genes that were previously well-established in *Drosophila*. If the regulatory mechanism of wing patterning in *Drosophila* (a holometabolous insect) applies to BPH (a hemimetabolous insect), it will be interesting to determine how *Nl*InR2 regulates LW development by exquisitely orchestrating the expression of wing-patterning genes. Another interestingly finding in this study was that depletion of *NlInR2* led to asymmetric vein patterning, but this phenotype was not observed in *NlInR2*[RNAi] BPHs [46]. Therefore, one speculative explanation for this different phenotypes could be that a basal level of *Nl*InR2 might be necessary for vein patterning in BPHs.

Notably, several case studies on wing polyphenism in Hemiptera insects and polyphonic horns in beetles indicated that the regulatory roles of multiple *InR*s on phenotypic plasticity might be insect lineage specific. In BPH and the linden bug *P. apterus*, two hemiptera insects, two *InR* paralogues had opposite roles in determining alternative wing morphs [28,46]. However, knockdown of *InR1* or *InR2* orthologues in the red-shouldered soapberry bug, *Jadera haematoloma* (Hemiptera: Rhopalidae), had a marginal role on wing-morph switching [59]. In the dung beetle *O. taurus*, knockdown of *O. taurus InR1* or *InR2* homologue had no significant effect on horn size [58], which is in marked contrast to shorted horns derived from knockdown of *InR1* in the rhinoceros beetle *T. dichotomus* [22].

The two BPH *InR* paralogues have evolved different functions for controlling alternative wing morphs, enabling BPHs to migrate or reside according to changes of heterogeneous environments. However, whether *InR* paralogues in other insects are involved in additional polyphenisms is still unknown. Although there remains much to be done, the current study on *NlInR2*-null mutants may provide a better understanding of the co-option of multiple *InR* paralogues in regulating multiple facets of life-history traits in insects. We believe that future experiments including comparative genomic analysis and functional genetic studies in more non-model insects will improve our understanding of the functional plasticity of multiple *InR* paralogues in insects.

## Materials and methods

### Insects

The BPH strain was originally collected from rice fields in Hangzhou (30˚16′N, 120˚11′E), China, in 2008. The *Wt*[SW] colony was purified by inbreeding for more than 13 generations, and used for genomic DNA sequencing and assembly [60]. All insects were reared at 26 ± 0.5˚C under a photoperiod of 16 h light/8 h dark at a relative humidity of 50 ± 5% on rice seedlings (strain: Xiushui 134).

### *In vitro* synthesis of Cas9 mRNA and sgRNA

The sgRNAcas9 algorithm [61] was used to search sgRNAs in the BPH genome using *NlInR2* sequence. The sgRNA was prepared as described previously [62], and then *in vitro* transcribed using the MEGAscript T7 high yield transcription kit (Thermor Scientific) according to the manufacturer's instructions. Cas9 mRNA was *in vitro* transcribed from plasmid pSP6-2sNLS-SpCas9 vector using the mMESSAGE mMACHINE SP6 transcription kit and Poly(A) tailing kit (Thermo Scientific).

## DNA typing for heterozygosity and homozygosity

Genomic DNA (gDNA) was isolated from whole BPH body or forewings to determine the heterozygous or homozygous genotypes of the BPHs. gDNA was extracted from one individual adult, as reported previously [63]. Briefly, one adult was homogenized in a 0.2-ml Eppendorf tube, followed by the addition of 50 μl of extraction buffer (10 mM Tris-HCl pH 8.2, 1 mM EDTA, 25 mM NaCl, 0.2 mg/ml proteinase K). The tubes were then incubated for 30 min at 37°C, followed by 2 min at 95°C to inactivate the proteinase K, and the supernatant solution was used directly as a template for PCR.

gDNA was isolated from forewings as reported previously [64], with slight modifications. Briefly, forewings were digested in 0.5 ml extraction buffer (0.01 M Tris-HCl, 0.01 M EDTA, 0.1 M NaCl, 0.039 M dithiothreitol, 2% sodium dodecyl sulfate, 20 μg/ml, pH 8.0) for 12h at 37°C, followed by 2 min at 95°C to inactivate proteinase K. The supernatant solution was then used directly as a template for PCR. PCR products spanning $NlInR2^{E4}$ and $NlInR2^{E5}$ sgRNA target sites were amplified from the extracted gDNA using primer pairs for E4-iF/E4-iR and E5-iF/E5-iR (S20 Table), respectively. The PCR products were then used for Sanger sequencing or subcloned into pEasy-T3 cloning vector (TransGen Biotech), and then single clones were picked for Sanger sequencing.

## Embryonic injection and crossing scheme

Embryonic injection was performed as described previously [65]. Pre-blastoderm eggs were dissected from rice sheaths within 1 h of oviposition, and microinjection manipulation was accomplished in the following 1 h. To perform cross-mating, a single male or female CRISPR/Cas9-injected $G_0$ adult was picked to mate with one $Wt^{SW}$ female or male to lay eggs for 10 days. Each $G_0$ adult was then homogenized to determine its genotype. Eggs ($G_1$ progeny) were reared to adulthood, and gDNA was then isolated from $G_1$ forewings for genotype determination. A single $G_1$ adult was picked to mate with one $Wt^{SW}$ adult to produce $G_2$ progeny, and a single $G_2$ adult was then allowed to mate with one $Wt^{SW}$ adult to produce homozygous mutants ($G_3$). The genotype for G2 and G3 generations were determined by dissecting forewings for gDNA isolation. A homozygous mutant population was derived by $G_3$ self-crossing.

## Developmental duration, glucose and triglyceride contents, fecundity, and adult longevity

Females were allowed to lay eggs for 2 h, and the hatched nymphs were then monitored every 12 h. Newly hatched 1st-instar nymphs ($n = 20$, 0–12 hAE) were collected, and each individual was raised separately in a glass tube. The developmental times of 1st-, 2nd-, 3rd-, 4th- and 5th-instar stages were monitored every 12 h. Glucose and triglyceride levels were measured in pooled 24h-adult females ($n = 15$) as reported previously [46]. Glucose levels were measured using glucose oxidase reagent (Sigma-Aldrich) according to the manufacturer's instructions. Triglyceride contents were quantified by enzymatic hydrolysis using the GPO Trinder method with a tissue triglyceride assay kit (Applygen Technologies), according to the manufacturer's instructions. The glucose and triglyceride contents were calculated based on three biological replicates. Adult longevity was determined by recording the mortality of newly emerged adult females (0–3 hAE, $n = 43$ for $WT^{SW}$ and $n = 42$ for $NlInR2^{E4}$) and males (0–3 hAE, $n = 52$ for $WT^{SW}$ and $n = 42$ for $NlInR2^{E4}$) every 12 h. To determine fecundity, newly emerged adult females (0–12 hAE) were collected for paired mating assays. Each female ($n = 20$) was allowed to match with two males in a glass tube. The insects were removed 10 days later and the laid eggs were counted under a Leica S8AP0 stereomicroscope.

## Starvation tolerance assay

Starvation tolerance assay was conducted as reported previously [47]. Briefly, newly emerged $Wt^{SW}$ and $NlInR2^{E4}$ adults (0–6 hAE, $n$ = 30) were collected, and provided with normal food for 24 h. Then, the BPHs were deprived of food and only provided water. Mortality was monitored every 8 h.

## Western blot analysis

Western blot analysis against Vg was performed as previously reported [66]. Briefly, ovaries were dissected from $Wt^{SW}$ and $NlInR2^{E4}$ females at 3 and 5 days after eclosion, and then homogenized in Pierce radioimmunoprecipitation assay buffer (Thermo Fisher Scientific). For immunoblot staining, equal amounts of protein were separated by sodium dodecyl sulphate polyacrylamide gel electrophoresis and then transferred to a polyvinylidene difluoride membrane (Millipore). The membrane was incubated with anti-Vg polyclonal rabbit antibody (1: 1000) for 1 h at room temperature (RT), followed by incubation with horseradish peroxidase conjugated goat anti-rabbit antibody (MBL life science) for 1 h at RT, The antibody against ß-actin was used as a loading control. The image of immunoreactivity was taken by the Molecular Imager ChemiDoc XRS+ system (Bio-Rad).

## Quantification of wing-cell number by qRT-PCR

Fifth-instar nymphs ($n$ = 150) collected at 3-, 24-, 36-, 48-, and 72-hAE were used for T2W and T3W dissection. gDNA was isolated from T2W and T3W, respectively, using a Wizard Genomic DNA Purification kit (Promega) according to the manufacturer's instructions. A primer pair (gDNA-F/R) targeting the single copy *Ultrabithorax* gene was used to quantify gDNA copy number in qRT-PCR assay. qRT-PCR was conducted using a CFX96TM real-time PCR detection system (Bio-Rad). Three independent biological replicates with three technical replicates were used for each sample.

## EdU staining of wing buds

Fifth-instar LW-destined $NlInR2^{E4}$ and SW-destined $Wt^{SW}$ nymphs at 24 hAE were microinjected with EdU (0.5 mM), and T2W and T3W were then removed from 48h-nymphs. Wing buds were fixed and dissected from the outer layer of chitin shell, as described previously [67]. Briefly, wing buds were pre-fixed by incubating in a cocktail solution (6 ml chloroform, 3ml ethanol, 1ml acetic acid) for 10 min at room temperature (RT), followed by fixation in FAA solution (5 ml 37% formaldehyde, 5 ml acetic acid, 81 ml ethanol, 9 ml $H_2O$) overnight at RT. After washing with methanol, outer layer of chitin shell was removed with forceps and the wing buds were fixed in 10% formaldehyde for 1 h at RT. After incubating with 1% Triton X-100 for 2h at RT, samples were used for EdU detection using a Click-iT EdU assay (Invitrogen) according to the manufacturer's instructions. Fluorescent images were acquired using a Zeiss LSM 810 confocal microscope (Carl Zeiss).

## RNAi-mediated gene silencing

The dsRNA synthesis and injection were performed as described previously [46]. Briefly, dsRNA primers targeting *NlInR1* (ds*NlInR1*) and *Gfp* (ds*Gfp*) were synthesized with the T7 RNA polymerase promoter at both ends (S20 Table). ds*NlInR1* and ds*Gfp* were synthesized using a MEGAscript T7 high yield transcription kit (Ambion) according to the manufacturer's instructions. Microinjection was performed using a FemtoJet microinjection system (Eppendorf). For morphological examination of ds*NlInR1*-treated BPHs, 4th-instar nymphs were

microinjected with ds*NlInR1* (100 ng each), and the morphology of the adults was photographed. To examine the antagonistic role of *NlInR1* in the context of *NlInR2*-null BPHs, 12h-5th-instar *NlInR2*E4 nymphs were microinjected with 150 ng ds*NlInR1* or ds*Gfp*, and the numbers of BPHs with alternative wing morphs were counted when the adults emerged. Adults (*n* = 5 for each of three replicates) at 24 h after eclosion were collected for examination of RNAi efficiency using qRT-PCR. The relative expression of *NlInR1* was normalized to the expression level of *rps15* with primers in S20 Table.

## Examination of indirect flight muscles by transmission electron microscope

Thoraxes were dissected from 24h-adults for transmission electron microscope, as in described previously [68]. Briefly, thoraxes were fixed in 2.5% glutaraldehyde overnight at 4˚C, followed by post-fixation in 1% osmium tetroxide for 1.5 h. Samples were sectioned and stained with 5% uranyl acetate followed by Reynolds' lead citrate solution. Sections were observed under a JEM-1230 transmission electron microscope (JEOL).

## Expression of wing-patterning genes in *NlInR2*-null mutants

Fifth-instar *NlInR2*E4 nymphs (*n* = 150) were collected at 24-, 48-, and 72-hAE. T2W and T3W were dissected and used for total RNA isolation. First-strand cDNA was synthesized from total RNA using HiScript QRTSuperMix (Vazyme). The qRT-PCR primers for *Nlen*, *Nlhh*, *Nldpp*, *Nlap*, *Nlser*, *Nlwg*, and *Nlvg* (S20 Table) were designed using Primer-Blast (https://www.ncbi.nlm.nih.gov/tools/primer-blast). The ribosomal protein gene *rps15* was used as an internal reference gene [69]. Statistical comparisons between samples were based on three biological replicates.

## RNA isolation, cDNA library preparation, and Illumina sequencing

Total RNAs were isolated using RNAiso Plus (TaKaRa) according to the manufacturer's protocol. The quality of the RNA was examined by 1% agarose gels electrophoresis and spectrophotometer (NanoPhotometer, Implen). RNA integrity was assessed using an RNA Nano 6000 Assay Kit with the the Agilent Bioanalyzer 2100 system (Agilent Technologies). A total of 1.5 μg RNA per sample was used for cDNA library construction using a NEBNext Ultra RNA Library Prep Kit for Illumina (NEB), following the manufacturer's recommendations.

## Read mapping and DEGs

After Illumina sequencing, clean reads were generated after removing adapter, polly-N, and low-quality reads from the raw data. All clean reads were aligned to the BPH reference genome using Hisat2 (v2.1.0). The aligned clean reads were coordinately sorted and indexed with Samtools (v1.9). Read counts and number of fragments per kilobase of transcript sequence per millions base pairs sequenced (FPKM) were calculated for each gene by using StringTie (v1.3.5). DEseq2 was used to screen DEGs with fold change $\geq 2$ and adjusted *P*-value $< 0.01$.

## GO enrichment analysis of differentially expressed genes

GO enrichment analysis of DEGs was performed using the online OmicShare tool (https://www.omicshare.com/tools/home/report/goenrich.html and https://www.omicshare.com/tools/Home/Soft/pathwaygsea).

## Statistical analysis

Results were analyzed using the two-tailed Student's *t*-tests, Pearson's Chi-Square test, and log-rank (Mantel-Cox) test. Data are presented as the mean ± standard error of the mean (mean ± s.e.m) for independent biological replicates. Significance levels are indicated as $P < 0.05$ (*), $P < 0.01$ (**), $P < 0.001$ (***), or $P < 0.0001$ (****).

## Supporting information

**S1 Fig. Alignment analysis of InR proteins.** Conserved domains of two ligand-binding loops (L1 and L2), a furin-like cysteine-rich region, three fibronectin type 3 region, a single trans-membrane (TM), and a tyrosine kinase region were indicated. The Cas9 cutting sites for *NlInR*^E4 and *NlInR2*^E5 mutants were indicated by arrowheads. *Pa*InR1a and *Pa*InR1b, the linden bug *Pyrrhocoris apterus* InR1 homologue (GenBank: KX087103.1 and KX087104.1). *Pa*InR2, *P. apterus* InR2 homologue (GenBank: KX087105.1). *Nl*InR1 and *Nl*InR2, the brown planthopper *Nilaparvata lugens* InR1 (GenBank: KF974333.1) and InR2 (GenBank: KF974334.1) homologue, respectively.
(TIF)

**S2 Fig. Morphology of compound eyes of *Wt*^SW and *NlInR2*^E4 adult females.**
(TIF)

**S3 Fig. Efficiency of RNAi-mediated *NlInR1* knockdown.** 48h-4^th-instar *Wt*^SW or 12h-5^th-*NlInR2*^E4 nymphs were microinjected with ds*NlInR1* or ds*Gfp*. BPHs (*n* = 5 for each of three replicates) at 24 h after adult eclosion were collected, and the *NlInR1* expression was examined in the context of for *Wt*^SW (**A**) and *NlInR2*^E4 (**B**) by qRT-PCR. The relative expression of *NlInR1* was normalized to the expression level of *rps15*. Statistical comparisons were performed using a two-tailed Student's *t*-test (**, $P < 0.01$ and ***, $P < 0.001$), and bars represent mean ± s.e.m.
(TIF)

**S4 Fig. Ovary morphology and vitellogenin (Vg) expression in ovary. (A)** Ovaries were dissected from *Wt*^SW, *NlInR1*^RNAi, and *NlInR*^E4 at 3 and 5 days after adult eclosion. *NlInR1*^RNAi was derived from 5^th-instar nymphs microinjected with ds*NlInR1*. **(B)** Western blotting assay of Vg in ovaries. Ovaries were immunoblotted with anti-Vg polyclonal antibody. The antibody against ß-actin was used as a loading control.
(TIF)

**S5 Fig. Comparative transcriptome analysis of *Wt*^SW, *NlInR2*^E4, and *NlInR1*^RNAi females at 12 h after adult eclosion. (A)** The number of differentially expressed genes (DEGs) in *NlInR1*^RNAi and *NlInR2*^E4 females compared to *Wt*^SW. **(B)** Top 20 enriched Kyoto Encyclopedia of Genes and Genomes (KEGG) pathways of common DEGs regulated by *NlInR1*^RNAi and *NlInR2*^E4. **(C)** Top 20 enriched KEGG pathways of DEGs specifically regulated by *NlInR1*^RNAi. **(D)** Top 20 enriched KEGG pathways of DEGs specifically regulated by *NlInR2*^E4.
(TIF)

**S1 Table. Data quality of RNA-seq.**
(XLSX)

**S2 Table. DEGs between InR2_2h4_T2W and WT_24h_T2W.**
(XLSX)

**S3 Table. GO terms of up-regulated DEGs between InR2_24h_T2W and WT_24h_T2W.**
(XLSX)

**S4 Table. GO terms of down-regulated DEGs between InR2_24h_T2W and WT_24h_T2W.**
(XLSX)

**S5 Table. DEGs between InR2_24h_T3W and WT_24h_T3W.**
(XLSX)

**S6 Table. GO terms of up-regulated DEGs between InR2_24h_T3W and WT_24h_T3W.**
(XLSX)

**S7 Table. GO terms of down-regulated DEGs between InR2_24h_T3W and WT_24H_T3W.**
(XLSX)

**S8 Table. DEGs between InR2_48h_T2W and WT_48h_T2W.**
(XLSX)

**S9 Table. GO terms of up-regulated DEGs between InR2_48h_T2W and WT_48h_T2W.**
(XLSX)

**S10 Table. GO terms of down-regulated DEGs between InR2_48h_T2W and WT_48h_T2W.**
(XLSX)

**S11 Table. DEGs between InR2_48h_T3W and WT_48h_T3W.**
(XLSX)

**S12 Table. GO terms of up-regulated DEGs between InR2_48h_T3W and WT_48h_T3W.**
(XLSX)

**S13 Table. GO terms of down-regulated DEGs between InR2_48h_T3W and WT_48h_T3W.**
(XLSX)

**S14 Table. GO terms of commonly regulated genes in wing buds of NlInR2E4 and WTSW.**
(XLSX)

**S15 Table. The expression level and annotation of genes in the GO term of cell cycle process.**
(XLSX)

**S16 Table. Data quality of RNA sequencing of females.**
(XLSX)

**S17 Table. Differentially expressed genes that were commonly regulated in NlInR1RNAi and NlInR2E4 females.**
(XLSX)

**S18 Table. Differentially expressed genes that were specifically regulated in NlInR1RNAi females.**
(XLSX)

**S19 Table. Differentially expressed genes that were specifically regulated in NlInR2E4 females.**
(XLSX)

**S20 Table. Primers used in this study.**
(XLSX)

**S1 Data. Transcriptomic analysis of wing buds in NlInR2E4 and WtSW BPHs.**
(DOCX)

**S2 Data. Transcriptomic analysis of NlInR2E4 and NlInR1RNAi female adults.**
(DOCX)

## Acknowledgments

We thank Dr. Dan-Ting Li for preparing Fig 3A and International Science Editing (http://www.internationalscienceediting.com) for language editing.

## Author Contributions

**Conceptualization:** Hai-Jun Xu.

**Data curation:** Wen-Hua Xue, Sun-Jie Chen, Jin-Li Zhang.

**Formal analysis:** Wen-Hua Xue, Jin-Li Zhang, Hai-Jun Xu.

**Funding acquisition:** Jin-Li Zhang, Hai-Jun Xu.

**Investigation:** Wen-Hua Xue, Nan Xu, Sun-Jie Chen, Xin-Yang Liu.

**Project administration:** Hai-Jun Xu.

**Resources:** Hai-Jun Xu.

**Supervision:** Hai-Jun Xu.

**Visualization:** Sun-Jie Chen.

**Writing – original draft:** Jin-Li Zhang, Hai-Jun Xu.

**Writing – review & editing:** Hai-Jun Xu.

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
