## [Decision Letter · Decision Letter 0]

1 Feb 2021

Dear Dr Xu,

Thank you very much for submitting your Research Article entitled 'Functionally independent evolution of a second insect insulin receptor gene' to PLOS Genetics.

The manuscript was fully evaluated at the editorial level and by independent peer reviewers. The reviewers appreciated the attention to an important problem, but raised some substantial concerns about the current manuscript. Based on the reviews, we will not be able to accept this version of the manuscript, but we would be willing to review a much-revised version. We cannot, of course, promise publication at that time.

If you decide to revise the manuscript for further consideration at PLOS Genetics, please aim to resubmit within the next 60 days, unless it will take extra time to address the concerns of the reviewers, in which case we would appreciate an expected resubmission date by email to plosgenetics@plos.org.

[LINK]

We are sorry that we cannot be more positive about your manuscript at this stage. Please do not hesitate to contact us if you have any concerns or questions.

Yours sincerely,

Subba Reddy Palli, Ph.D.

Associate Editor

PLOS Genetics

Kirsten Bomblies

Section Editor: Evolution

PLOS Genetics

Reviewer's Responses to Questions

**Comments to the Authors:**

Reviewer #1: Major opinion and suggestion.

The main argument of this manuscript is that in BPH, InR1 and InR2 are functionally different in largely two points; 1. Life history aspect, and 2. Wing polymorphism. InR2 knockout affected nymph development duration and fecundity. This manuscript used previous report [reference 28] where phenotype of InR1 knockdown by RNAi is demonstrated, to compare with InR2 knockout phenotype. I think for this manuscript to gain more originality and novelty, it should include more detailed mechanism of how each InR1 and InR2 work differently. This manuscript did a good job in finding distinct phenotypes of InR2 knockout, so there should be distinct mechanism of InR2 gene action compared to InR1 which leads to different functions. In terms of wing polymorphism, detailed mechanism of InR1 and InR2 is already reported [reference 28], and this manuscript reemphasized this previous argument by showing increased cell number and increased mRNA expression coding for cell cycle and DNA replication related pathway in InR2 knockout wing disc. Evolution of insect insulin receptor in different insect species is already investigated in reference 27. Therefore, I am suggesting that focusing more on comparison of life history aspect of each InR1 and InR2 knockout (or knockdown), and identifying molecular mechanisms to explain this consequence will improve novelty of this manuscript. Is InR2 able to transduce ILP signal? Does InR2 have different ILP ligand than InR1? What is the function of InR2 in ovary or fat body to regulate fecundity? I believe answering such questions will further support the title of this manuscript ‘Functionally independent evolution of a second insect insulin receptor gene’.

Minor questions.

1. InR2 where is the functional domain? Since the sgRNA target exon 4 and 5, is there any possibility that truncated protein is still active?

2. Fig 1. Scale bar missing in in D and E. Especially in Fig 1E, I assume the size of WT and NlInR2E4 is different.

3. Fig 4. Fifth instar NlInR2E4 nymphs at 12 hAE were microinjected… But in the text in line 187, it is mentioned that 24h5th-instar was microinjected. Which one is true?

4. The title is too ambitious. It would be more informative if title included the name of the species studied (BPH).

5. InR2 mutant has increased cell cycle and number in wing disc. Is this effect restricted to wing disc? If this effect is wing disc specific, what is the explanation for this phenomenon?

Reviewer #2: SUMMARY: Thank you for giving me the opportunity to review this manuscript by Xue et al. This manuscript examines the potential functional significance of a second insulin receptor in the brown plant hopper (BPH). Previous work using RNAi had established that InR1-RNAi caused the formation of miniature short-winged morphs (SW) with extended life span yet decreased fertility and compromised carb and lipid metabolism. In contrast InR2RNAi had no such consequences and instead shifted development to long winged morphs (LW). This was the first indication of functional divergence between InR paralogs in an insect. The existence of multiple paralogs in insects is generally common, though absent in Drosophila and C. elegans, and hence much understudied. In the present ms the authors expand on these earlier findings by employing CRISPR to create two different null mutants for InR2. They document that (a) InR2 null mutants mad minimal effect on fuel metabolism and life span (in contrast to what has been documented in flies and worms) (b) induce wing growth through (c) increased cell proliferation associated with (d) the upregulation of genes linked to cell growth and proliferation and (e) wing patterning genes. More generally the manuscript deepens understanding of the functional divergence between two InR paralogs in plant hopper development, and is likely to motivate many follow up studies in other non-model insect species.

SUMMARY EVALUATION: The ms is well written, thorough, and detailed. It addresses an important topic - because only a single InR exists in the by far most studied insect, our understanding of potential functional divergences of multiple InR paralogs in insect development and evolution is very modest, even though having more than one InR is typical of almost all insects. While the ms does not document evidence for functional divergence between paralogs in BPH for the first time, it significantly expands our understanding thereof, using technologies (CRISPR) that mark a major advancement in methodology for insect evo devo in hemimetabolous insects. All combined I view PLoS Genetics as an appropriate venue for this ms. Below I detail several major and minor comments that I hope the authors will find of use.

SIGNED: Armin Moczek

MAJOR COMMENTS:

It was not clear to me (a) what the variation in body size was that existed among individuals within treatment groups, and whether efforts were made to correct for that variation in the statistical analyses. For example, larger female adults generally produce more eggs in most insects, so if WtSW animals are on average a bit bigger than NIInR2E4 animals (as seems to be the case from Fig. 2 B) then the difference in fecundity reported in figure 2 may just be a function of body size. Same goes for the metabolic data: glucose and triglyceride trend non-significantly to be less abundant in Wt animals, but if their relatively greater size is factored in this difference may become significant.

The authors focused on wings predominantly, for good reasons. But looking at the animals presented in Fig 1 D I immediately wonder what happened to other traits, including traits that already exist in nymphs (say eyes) and those that don't (like genitalia). If the authors want to hold this for another publication I can understand but if not it would make a possibly really nice addition

Fig 5: could the authors clarify if the go term analysis was contrasted to some kind of null expectation? As it is the results are used to support the hypothesis that InR2 regulates genes related to cell proliferation. Alternatively, this result could be expected by chance. Could the authors simulate what go term distributions they would get if drawing 657 genes at random from the BPH genome, then juxtapose that to their measured GO term frequencies?

lines 270 onwards report expression changes in wing patterning genes following InR2 null mutations. This should probably be moved into the results section.

line 277: I agree that much more work is needed to examine InR paralog functions in other insects. In this section I would encourage the authors to cite and briefly discuss the work of Sofia Casasa on Onthophagus beetles published in Proceedings B. While she did not find strong evidence for functional divergence, she did examine both paralogs independently and I think that is worth mentioning. This then would allow the authors to explore two more points: (a) that sub/neofunctionalization may be lineage specific, possibly common in Hemiptera or Hemimetabola but not Holometabola and why that might be and (b) contrast Sofia's finding to their reference (22): InR RNAi affected horn formation in one subfamily of scarab beetles (Emlen et al) but the other (Casasa). Here already there is a solid hint that among subfamilies in teh same insect order InR function may diverge quite significantly.

MINOR COMMENTS

line 118: symmetrical patterning of wings: I suggest rewriting this and to focus on wing venation. When I read this I expected an analysis of symmetry, but the phenotypes that are described all focus on reduction in parts of the venation system

line 158: ...increase wing cell number". Mentioning number specifically is important here I think because when I read that I thought wow specific cells in the wing get bigger

like 183: what is IFM? In general be mindful of overloading your reader with too many abbreviations. This one in particular though does not seem defined, at least I could not find it and after a while gave up so the hole section is lost on me.

WRITING SUGGESTIONS

The ms is very well written but on occasion some clarifications or changes in word choice might help. Here are some suggestion, please use what you like

Line 1: Consider changing "Functionally independent evolution of..." to "Neofunctionalization of ..."

25: However, most insects possess additional copies of InR, yet the functional significance, if any, of alternate InRs is unknown

30: "but differed greatly" to "but revealed distinct regulatory roles in"

48: such

54: "possesses" instead of "had"

74: "in higher organisms" to "in mice and humans".

56: ...may have functionally diversified in ways more complex than..

123: "null mutants develop into viable LW morphs"

also consider not using too many abbreviations in titles or subtitles

152: ...females showed a ~27% reduction in fecundity...

195: These results

221: our understanding of their functions remains limited

268: To gain insights OR To look for insights

.

Reviewer #3: The manuscript describes studies that assess whether roles identified for an insulin receptor (InR1) in previous work are shared or differ for a second insulin receptor (InR2) encoded in the genome of the brown plant hopper (BPH). Using the CRISPR/Cas9 system, the authors disabled expression of InR2 in BPH and found that its absence was not lethal and favored development of long winged adults, unlike the control and InR1 RNAi treated individuals that became short-winged adults. InR2 mutants also took longer to develop and had reduced fecundity relative to control or InR1 RNAi individuals. Other related aspects of life history, metabolism, and gene expression were also examined and compared for the different treatment groups, either produced here or in previous work. The experimental design is appropriate, and the manuscript is well written and organized with lots of data for different but related biological aspects. The results and inferences are based on the phenotypic outcome of reverse genetics solely but do provide important insight into possible differences in InR1 and InR2 regulation of developmental and physiological processes in this up and coming model insect.

Below are questions or suggestions for the authors to address.

- Comparison/contrast of the pleiotropic regulation extensively characterized for the single InR in Drosophila and C. elegans to what little is known for the two InRs in BPH is repetitive and largely meaningless. More background information should be provided about other insect groups with two or more InRs and what has been discerned about their expression, signaling, and function. More relevant would be an examination of the extensive literature on the endocrine regulation of long/short wing cricket morphs and other hemimetabolous insects more closely related to BPH than Drosophila, which is a highly derived insect anomaly. More information especially about the structural and functional similarity of two InRs in BPH and Pyrrhocoris is warranted.

- A bit more context should be provided in the Introduction or Discussion about the number and type of insulin-like peptides, ligands for the InRs, and relevant downstream signaling elements encoded in BPH that may differentially affect the regulation/function of the two InRs.

- Efficacy of the InR1 RNAi should be shown for the experimental sets described in this work. The same should be shown for the CRISPR/Cas9 InR2 mutants – are any full length InR2 transcripts detected by PCR in whole insects? Perhaps it is not completely disabled in all tissues, which may affect the interpretation of the results.

- The GenBank accession number should be provided for the InR1.

- There is no explanation in the Methods section (lines 347-354) how “relative cell number” was obtained/converted for the data points in Fig. 3B from the Ultrabithorax qRT-PCR data. Some procedure/formula with a reference is needed.

- The punctuation or sentence structure in line 189 should be changed to separate the mutant annotations for better understanding of what is being conveyed.

- In the Discussion, expression of genes involved in wing patterning is compared for InR2 mutant and wild BPH – lines 262-276. The data presented in S1 Fig appear to be good enough and provide depth to the work and should be included in the Results in my opinion.

- Line 29 – “NlInR2 phenocopied…” makes no sense. Instead of using “phenocopied” (is it even a real word?), provide a more complete description that will aid the reader’s understanding.

- What defines a “canonical” InR in any organism? Don’t think it is an appropriate modifier any place used in the text.

- Line 78 – “two types of receptors are unified into a single InR in the fly” – there is no evidence for this especially since the single InR in the fly was there long before the InR and IGFR showed up in vertebrates.

- Line 146 – “bona fide” is an adjective or noun and not used correctly here.

- Line 158 and another place – “To look insight into…” – drop “insight” or reword as “to gain insight”.

- Lines 203 – 215 – The discussion of the results is repetitive and confusing.

- The title is too vague and should include information about the organism.

**Have all data underlying the figures and results presented in the manuscript been provided?**

Reviewer #1: Yes

Reviewer #2: None

Reviewer #3: Yes

PLOS authors have the option to publish the peer review history of their article (what does this mean?). If published, this will include your full peer review and any attached files.

Reviewer #1: No

Reviewer #2: **Yes: **Armin Moczek

Reviewer #3: No

---

## [Decision Letter · Decision Letter 1]

21 May 2021

Dear Dr Xu,

Thank you very much for submitting your Research Article entitled 'Neofunctionalization of a second insulin receptor gene in the wing-dimorphic planthopper, Nilaparvata lugens' to PLOS Genetics.

The manuscript was fully evaluated at the editorial level and by independent peer reviewers. The reviewers appreciated the changes in response to their comments but reviewer #2 identified some concerns about statistical analysis of RNAi data that we ask you address in a revised manuscript

We therefore ask you to modify the manuscript according to the review recommendations. Your revisions should address the specific points made by reviewer #2.

[LINK]

Yours sincerely,

Subba Reddy Palli, Ph.D.

Associate Editor

PLOS Genetics

Kirsten Bomblies

Section Editor: Evolution

PLOS Genetics

Reviewer's Responses to Questions

**Comments to the Authors:**

Reviewer #1: The questions from previous review are addressed well. I think the data in the manuscript and additional experiments listed in response to review are convincing enough to show that NlInR1 and NlInR2 have distinct functions in BPH life history aspect. Experiments listed in response to review, especially the RNAseq data from adult female (Experiment #2) can be included in the supplementary file, unless the author want to use the data for other publication. It may not be the scope of this manuscript, but I think the point in line 328 (NlInR2 may serve as a negative regulator of NlInR1, as proposed previously [46]) is an important hypothesis worth testing. If there is any BPH cell line available, and if increased phosphorylation of NlAkt can be achieved by overexpressing NlInR1, an experiment can be designed to co-express NlInR1 and NlInR2 to see whether there is decrease in P-NlAkt compared to NlInR1 overexpression only. Since in reference [46], it is shown that NlInR and NlInR2 can bind each other [46: Figure2H], checking whether overexpressed NlInR2 can inhibit activity of NlInR1 mediated downstream insulin signaling can lead to interesting finding.

Below are some minor suggestions.

In line 183

wing veins in forewings and hindwings, leading to a single individual with different wing patters at both sides of its body

In 213

.. our findings indicate that NlInR2 resembles …

In 257

NlInR1 knockdown could antagonize the effects of NlInR2 knockout on wing and indirect flight muscles development

In 282

morphogens hh, dpp, and wg

Reviewer #2: I think the authors have done a thorough and thoughtful set of revisions, addressing all my comments and I think most of those of the other reviewers. My only remaining suggestion would be to redo the statistical analysis of the effects of InR knockdowns on fecundity and other traits. Rather than using pairwise comparisons I think the authors should use a simple analysis of variance to be able to describe how much of e.g. changes in fecundity can be attributed to body size alone and how much to treatment. Right now the authors are guessing that some of these effects are negligible, this way they would know for sure. Further, they would be able to identify possible interaction effects, which right now they have no way of seeing. Any statistical consulting center should be able to help with this. Other than that minor issue the ms is in fine shape in my opinion.

Reviewer #3: The revised manuscript is much improved and a clearer case made for this important contribution. The authors satisfactorily addressed all of my suggestions and concerns.

**Have all data underlying the figures and results presented in the manuscript been provided?**

Reviewer #1: Yes

Reviewer #2: Yes

Reviewer #3: Yes

PLOS authors have the option to publish the peer review history of their article (what does this mean?). If published, this will include your full peer review and any attached files.

Reviewer #1: No

Reviewer #2: **Yes: **Armin Moczek

Reviewer #3: No

---

## [Editor Report · Decision Letter 2]

9 Jun 2021

Dear Dr Xu,

We are pleased to inform you that your manuscript entitled "Neofunctionalization of a second insulin receptor gene in the wing-dimorphic planthopper, Nilaparvata lugens" has been editorially accepted for publication in PLOS Genetics. Congratulations!

Yours sincerely,

Subba Reddy Palli, Ph.D.

Associate Editor

PLOS Genetics

Kirsten Bomblies

Section Editor: Evolution

PLOS Genetics

Comments from the reviewers (if applicable):

**Data Deposition**

http://datadryad.org/submit?journalID=pgenetics&manu=PGENETICS-D-20-01928R2

**Press Queries**

---

## [Editor Report · Acceptance letter]

23 Jun 2021

PGENETICS-D-20-01928R2 

Neofunctionalization of a second insulin receptor gene in the wing-dimorphic planthopper, Nilaparvata lugens 

Dear Dr Xu, 

We are pleased to inform you that your manuscript entitled "Neofunctionalization of a second insulin receptor gene in the wing-dimorphic planthopper, Nilaparvata lugens" has been formally accepted for publication in PLOS Genetics! Your manuscript is now with our production department and you will be notified of the publication date in due course.

With kind regards,

Katalin Szabo

PLOS Genetics

On behalf of:
